# Comprehensive evaluation of high-oleic rapeseed (*Brassica napus*) based on quality, resistance, and yield traits: A new method for rapid identification of high-oleic acid rapeseed germplasm

Tao Chang[1,2], Junjie Wu[1,2], Xuepeng Wu[1,2], Mingyao Yao[1,2], Dongfang Zhao[1,2], Chunyun Guan[1,2,3], Mei Guan[1,2,3]*

**1** Hunan Branch of National Oilseed Crops Improvement Center, Changsha, China, **2** College of Agriculture, Hunan Agricultural University, Changsha, China, **3** Southern Regional Collaborative Innovation Center for Grain and Oil Crops in China, Changsha, China

* gm7142005@hunau.edu.cn

**Data Availability Statement:** All relevant data are within the manuscript and its Supporting information files.

## Abstract

To scientifically evaluate and utilize high-oleic acid rape germplasm resources and cultivate new varieties suitable for planting in the Hunan Province, 30 local high-oleic acid rape germplasms from Hunan were used as materials. The 12 personality indices of quality, yield, and resistance were comprehensively evaluated by variability, correlation, principal component, and cluster analyses. The results of variability showed that except for oleic acid, the lowest coefficient of variation was oil content, which was 0.06. Correlation analysis showed that oil content was positively correlated with main traits such as yield per plant and oleic acid, which could be used in the early screening of high-oleic rape germplasm. The results of principal component analysis showed that the 12 personality indicators were integrated into four principal components, and the cumulative contribution rate was 62.487%. The value of comprehensive coefficient 'F' was positively correlated with the first, second, and fourth principal components and negatively correlated with the third principal component. Cluster analysis showed that 30 high-oleic rape germplasms could be divided into four categories consisting of 9 (30%), 6 (20%), 7 (23%), and 8 (27%) high-oleic rape germplasms, each with the characteristics of "high disease resistance", "high yield", "high protein", and "more stability". This study not only provides a reference basis for high-oleic rape breeding but also provides a theoretical basis for their early screening.

## Introduction

Oilseed rape (*Brassica napus* L.) is the largest oil crop in China. The output of rapeseed oil accounts for more than 50% of the output of domestic oil crops, and plays an important role in

**Funding:** China Agriculture Research System of MOF and MARA(CARS-13)" and "Hunan Agriculture Research System of DARA (Xiangnongfa[2019]No.105).

**Competing interests:** The authors have declared that no competing interests exist.

maintaining the national edible oil supply security strategy [1]. With the improvement in living standards, people's demand for enhanced edible vegetable oil quality is increasing year by year. In the past, linoleic and linolenic acid were considered the essential fatty acids for the human body; hence, they have attracted much attention [2]. However, recent studies have shown that oleic acid is the more important essential fatty acid [3]. High oleic acid rape oil not only has high smoke point, but also protects cardiovascular health [4], which is favored by the majority of consumers.

The research teams at the Chinese Academy of Engineering and Hunan Agricultural University, have formulated the national standard for high-oleic rapeseed (NY/T 3786–2020), and its production has officially become an industrialized system. At present, the strategic goal for oilseed rape production is to further improve the quality and healthcare properties while ensuring or maintaining a high yield [5, 6]. However, the production of high-oleic rape has been affected by problems such as reduced yield, prolonged growth period, low economic benefit, and limited promotion, which are affected by the breeding process [7]. Previous relevant studies mostly focused on unilateral character improvement. For example, Guo et al. [8] comprehensively analyzed the yield characters of 130 *Brassica napus*, screened 12 materials with excellent comprehensive characters and selected 6 principal components that can be used for evaluation. Zhu et al. [9] identified 49 *Brassica napus* germplasm resources for drought resistance and selected resources with different drought tolerance. Failure to consider scientific and effective means to associate oleic acid with other characteristics, resulting in the slow progress of variety improvement and becoming a bottleneck in the development of the rape industry.

Crop quality, yield, and resistance are quantitative traits controlled by multiple genes [10]. Due to the correlation between different forms, it is difficult to analyze them comprehensively. Multivariate statistical analyses, including variation, correlation, principal component, and cluster analyses, are widely used in the comprehensive evaluation and analyses of oilseed rape traits [11]. Principal component analysis can combine multiple interrelated quantitative trait indices into a few independent principal components by reducing the dimension [12]. Cluster analysis can effectively classify all materials and preliminarily judge the genetic relationship between different varieties [13].

In recent years, researchers have carried out extensive studies related to the comprehensive evaluation of rape varieties, providing a reference basis for breeding rape varieties [14–16]. However, to the best of our knowledge, there are no relevant studies available on high-oleic rape. There are few reports on the comprehensive evaluation of yield and agronomic traits of high-oleic acid rape using multivariate statistical methods.

Germplasm resources are an important basis for biological research and genetic breeding of oilseed rape [17]. Varieties which are developed and evaluated in local conditions are often better suited to local natural or cultivation conditions than foreign varieties and have excellent traits or genes to adapt to the local environment [18]. This research was conducted in the large rapeseed producing area of Hunan Province. Transforming traditional rapeseed oil into high oleic oil could provide substantial economic and nutritional benefits to 61 million people in the province. This study collected the natural high-oleic acid rape resources in the Hunan Province and comprehensively evaluated the quality, yield, and resistance traits. These traits which are conducive to the introduction and domestication of local varieties and the evaluation and utilization of excellent trait genes to lay a foundation for the cultivation of new high-oleic acid varieties and the sustainable development of the industry.

## Materials and methods

### Overview of the experimental field

The experimental site was located in the Yunyuan Base of the Hunan Agricultural University, Changsha, Hunan, China (113˚070' E, 28˚180' N), with a humid subtropical monsoon climate. During the experiment(September 2020 to May 2021), the cumulative rainfall was 158.3 mm, and the average temperature was 16.2˚C. The experimental field implemented a "rice-oilseed" rotation.

### Test methods

Ten plants for each variety were selected, and the whole plant was harvested to detect the quality and yield traits when rapeseeds plants were mature. The quality traits of the oilseed rape samples which were measured included crude protein and fatty acid composition. The crude protein was determined using near-infrared analysis [19]. And fatty acid composition, percentage of oleic, palmitic, stearic, linoleic, linolenic, and erucic acid in total fatty acids, was determined using gas chromatography [20]. The yield traits of samples included oil content, 1000-grain weight and yield per plant. The oil content was measured using the residue weight method [21]. The1000-grain weight and yield per plant was were conducted as described by Zhao [22]. The resistance traits of samples were detected on day 28 after the pod appearance and included incidence rates of *Sclerotinia sclerotiorum* and viral diseases, which were performed as described by Wei [23].

### Materials

The test material consisted of 30 *Brassica napus* plants with high oleic acid content. Among them, materials 1–4 were approved varieties of *Brassica napus* with high oleic acid, respectively, named Xiangyou 708, Xiangyou 710, Xiangzayou 991, and Xiangzayou 992, while materials 5–30 were stable inbred lines of *Brassica napus* with high oleic acid. The origin of these high-oleic rapes was in Chasha, Hunan province. These materials were provided by the Hunan branch of the National Oil improvement Center.

### Date analyses

Microsoft Excel 2003 was used for statistical analyses. SPSS 24.0 was used for cluster, correlation, and principal component analyses. The comprehensive score $F$ of principal components and phenotypic traits were calculated and then combined with stepwise regression analysis to screen the evaluation indices of comprehensive characters. The boxplot of traits was drawn using SPSS 24.0. The membership function analysis, genetic diversity index, and stability evaluation of germplasm resources were conducted as described by Kumar [24].

## Results and analysis

### Frequency distribution and diversity of traits in high-oleic rapes

The traits of quality, yield, and resistance showed (Fig 1) that the percentages of oleic acid, palmitic acid, stearic acid, linoleic acid, linolenic acid and erucic acid in the figure are the percentages of total fatty acids. The percentages of protein and oil content in the figure are the percentages of whole seed.

The results showed that the oleic acid content of high-oleic rapeseed was evenly distributed between 75–84%. The palmitic acid content was mostly concentrated in 3–5% with a frequency of 93%. The range of stearic acid was small, showing values between 1–2%. The linoleic acid in

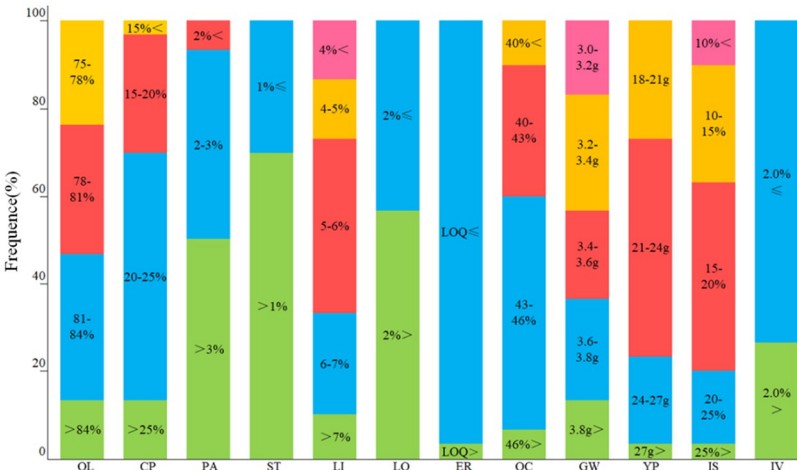

**Fig 1. Variation distribution of qualitative traits of high-oleic rape germplasms.** OL: Oleic, CP: Crude protein, PA: Palmitic acid, ST: Stearic acid, LI: Linoleic acid, LO: Linolenic acid, ER: Erucic acid, OC: Oil content, GW: 1000-grain weight, YP: Yield per plant, IS: Incidence rate of *Sclerotinia clerotiorum*, IV: Incidence rate of viral diseases. The percentages of oleic acid, palmitic acid, stearic acid, linoleic acid, linolenic acid and erucic acid are the percentages of total fatty acids. The percentages of oil content and crude protein were obtained as their percentages in whole seed. The percentages of *Sclerotinia sclerotiorum* and viral diseases were obtained as their incidence rate in these plant materials. The same as below. LOQ: Erucic acid standard of "double low" rapeseed oil, 3%.

high-oleic rape was mainly less than 21% with 86% frequency. The linolenic acid was primarily concentrated in 4–8% with a frequency of 86%. Erucic acid was not present in high-oleic rape except in some cases. Protein mostly showed a concentration in the range of 15–25% with a frequency of 83%. The oil content of high-oleic rape was largely more than 40% but not more than 46% with 83% frequency, which is a medium oil production level. The 1000-grain weight was evenly distributed between 3–4 g and yield per plant was almost concentrated in 18–27 g with 96% frequency. The incidence rate of *Sclerotinia* was higher than that of viral diseases in high-oleic rapes, but it was lower than 25%.

## Variation analyses of quality, yield, and resistance traits of different high-oleic rape germplasms

The 12 traits of 30 high-oleic rapeseed showed different degrees of variation, as shown in Table 1. Except for erucic acid, the coefficients of variation of other traits were low, indicating that the high-oleic rapeseed germplasms in Changsha tended to be stable. However, the standard deviation of some traits such as oleic acid, crude protein, linoleic acid, and the incidence rate of *Sclerotinia sclerotiorum* is high. In other words, within a specific range, these traits showed diversity. The results showed that the maximum *Sclerotinia sclerotiorum* concentration in high-oleic rape was more than three times the minimum value, and the average was 16.42%, with a range of 8.3%-29.1%.

## Correlation analyses of quality, yield, and resistance traits of different high-oleic rape germplasms

The correlation analyses of quality, yield, and resistance trait indices of high-oleic rape germplasms (Fig 2) showed different degrees of correlation among the traits. The results showed that oleic acid had significantly high positive correlations with stearic acid, significant positive correlations with crude protein and linoleic acid, and significant negative correlations with

**Table 1. Variation coefficients of traits in different high-oleic rapeseed.**

| Trait | Min | Max | AV | SD | CV |
|---|---|---|---|---|---|
| **Quality trait** | | | | | |
| **OL/%** | 75.42 | 85.21 | 80.57 | 3.00 | 0.04 |
| **CP/%** | 12.27 | 27.74 | 21.26 | 3.45 | 0.16 |
| **PA/%** | 1.93 | 3.70 | 2.95 | 0.39 | 0.13 |
| **ST/%** | 0.81 | 1.58 | 1.22 | 0.25 | 0.21 |
| **LI/%** | 3.37 | 7.71 | 5.50 | 1.18 | 0.22 |
| **LO/%** | 1.34 | 2.89 | 2.08 | 0.45 | 0.22 |
| **ER/%** | 0 | 0.51 | 0.04 | 0.10 | 2.84 |
| **Yield trait** | | | | | |
| **OC/%** | 35.52 | 48.23 | 42.90 | 2.71 | 0.06 |
| **GW/g** | 3.08 | 3.95 | 3.47 | 0.26 | 0.07 |
| **YP/g** | 18.36 | 27.52 | 22.38 | 2.59 | 0.12 |
| **Resistance trait** | | | | | |
| **IS/%** | 8.3 | 29.1 | 16.42 | 4.80 | 0.29 |
| **IV/%** | 1.1 | 2.6 | 1.71 | 0.38 | 0.22 |

Min: Minimum, MAX: Maximum, AV: Average, SD: Standard deviation, CV: Coefficient of variation; Note: The percentages of oleic, palmitic, stearic, linoleic, linolenic and erucic acid were obtained as percentages of these components in total fatty acid.

erucic acid, oil content, and yield per plant. Crude protein exhibited a significantly high positive correlation with linoleic acid, a significant positive correlation with linolenic acid, and significant negative correlations with erucic acid, yield per plant, and incidence rate of viral diseases. Palmitic acid had a significantly high positive correlation with linoleic acid, significant positive correlations with yield per plant and incidence rate of viral diseases, and a significantly high negative correlation with erucic acid. Stearic acid showed significant positive correlations with linoleic acid and incidence rate of *Sclerotinia sclerotiorum*, and significant

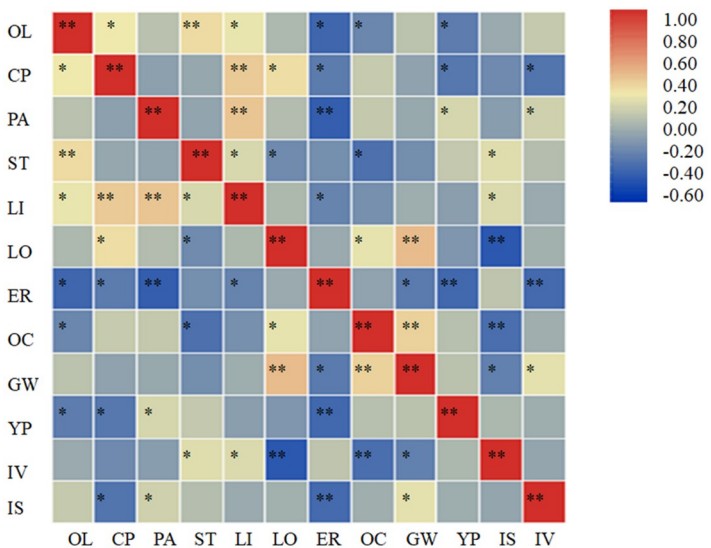

**Fig 2. Correlation analyses of traits of different high-oleic rape germplasms.**

negative correlations with linolenic acid and oil content. Linoleic acid had a significant positive correlation with the incidence rate of *Sclerotinia sclerotiorum* and a significant negative correlation with erucic acid. Linolenic acid showed a significantly high positive correlation with 1000-grain weight, a significant positive correlation with oil content, and a significantly high negative correlation with the incidence rate of *Sclerotinia sclerotiorum*. Erucic acid had significant negative correlations with 1000-grain weight, yield per plant, and incidence rate of viral diseases. Oil content displayed a significantly high positive correlation with 1000-grain weight and significant negative correlations with the incidence rate of viral diseases. The 1000-grain weight exhibited significant positive correlations with the incidence rate of viral diseases and a significant negative correlation with the incidence rate of *Sclerotinia sclerotiorum*.

## Principal component analyses (PCA) of quality, yield, and resistance traits of different high-oleic rape germplasms

The PCA results of quality, resistance, and yield characteristics of high-oleic rape are shown in Fig 3. The results showed that most varieties were grouped together, indicating that the data were reliable and stable (Fig 3A). The PCA data analyses showed that the eigenvalues of the first four principal components were $\lambda \geq 1$. The cumulative contribution rate of the first four principal components was 62.487%, which represented 62.487% of the traits of 30 high-oleic rape germplasms, and could be evaluated by the first four principal components.

The variance contribution rate of the first principal component was 19.506%. The first principal components were mainly linolenic acid, 1000-grain weight, erucic acid, *Sclerotinia sclerotiorum* incidence rate, and oil content, and their load values were 0.666, 0.635, −0.550, −0.505, and −0.500, respectively. The variance contribution rate of the second principal component was 18.009%. The main character indices determining the two principal components were linoleic, stearic, and oleic acid, and their load values were 0.649, 0.642, and 0.597, respectively. The variance contribution rate of the third principal component was 14.538%. The third

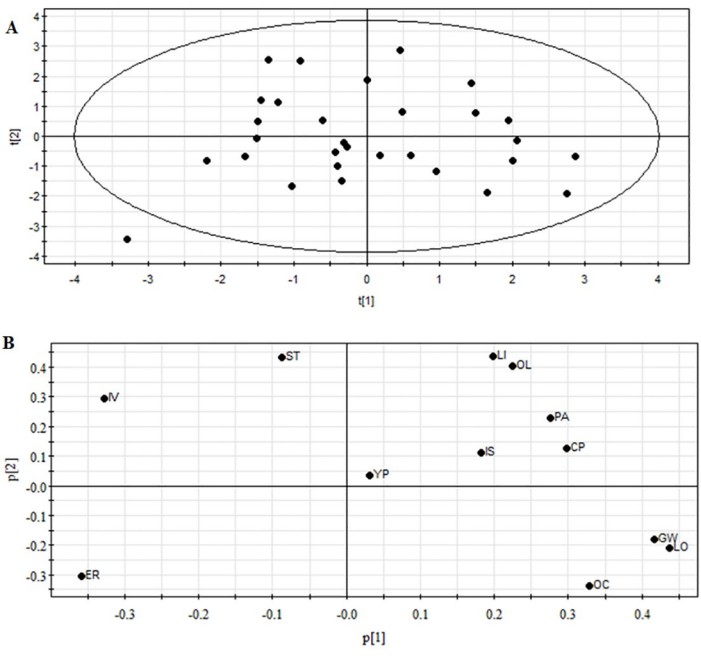

**Fig 3. Principal component analyses (PCA) of varieties (A) and traits (B).**

principal components were mainly crude protein, yield, and virus incidence rate, and their load values were −0.700, 0.683, and 0.514, respectively. The variance contribution rate of the fourth principal component was 10.435%. The fourth principal components were mainly palmitic acid and the incidence rate of viral diseases, and their load values were 0.506 and −0.505, respectively. Moreover, each trait was in a different interval (Fig 3B) in cluster analysis, which could be further used to establish a model for comprehensive evaluation.

## Comprehensive evaluation of quality, yield, and resistance traits of different high-oleic rape germplasms

To eliminate the influence of data dimension in the comprehensive scoring of 30 high-oleic rape germplasms, the average value of each trait index was standardized using SPSS. According to the standardized value and load matrix, the linear equation of each principal component score was obtained and then the variance contribution rate corresponding to the four principal components was used as the weight coefficient to establish a comprehensive evaluation model for comprehensive evaluation score. The linear equation and comprehensive evaluation model of the four principal component scores were as follows:

$$F_1 = 0.341X_1 + 0.454X_2 + 0.421X_3 - 0.137X_4 + 0.301X_5 + 0.666X_6 - 0.550X_7 + 0.500X_8 - 0.635X_9 + 0.045X_{10} - 0.505X_{11} + 0.276X_{12}$$

$$F_2 = 0.597X_1 + 0.192X_2 + 0.340X_3 + 0.642X_4 + 0.649X_5 - 0.303X_6 - 0.440X_7 - 0.492X_8 - 0.260X_9 + 0.055X_{10} + 0.436X_{11} + 0.171X_{12}$$

$$F_3 = -0.248X_1 - 0.700X_2 + 0.333X_3 + 0.070X_4 - 0.255X_5 - 0.277X_6 - 0.362X_7 + 0.112X_8 + 0.234X_9 + 0.683X_{10} + 0.085X_{11} + 0.514X_{12}$$

$$F_4 = -0.431X_1 + 0.205X_2 + 0.506X_3 - 0.272X_4 + 0.369X_5 - 0.093X_6 - 0.021X_7 + 0.226X_8 - 0.321X_9 + 0.365X_{10} + 0.086X_{11} - 0.505X_{12}$$

$$F = (19.506F_1 + 18.009F_2 + 14.538F_3 + 10.435F_4)/62.487$$

Where, $X_1 \sim X_{12}$ represent oleic acid, crude protein, and palmitic, stearic, linoleic, linolenic, and erucic acid, oil content, 1000-grain weight, yield per plant, and the incidence rate of *Sclerotinia sclerotiorum* and viral diseases, respectively.

The comprehensive score of each high-oleic acid rapeseed germplasm was calculated and ranked using the above model (Table 2). The results showed that in addition to the new varieties that have been put into production, 6 materials had a high comprehensive evaluation, of which germplasm No. 14 ranked first or can be used to register new varieties. The germplasm No. 5, No. 26, No. 15, No. 6, and No. 9 showed high comprehensive scores and could be used as reserves of high-oleic acid rape germplasm resources.

The scores of the four principal components were used as ordinate, and the comprehensive ranking was used as abscissa. The trends of map were observed, and the results (Fig 4) showed that principal components 1, 2, and 4 showed a downward trend ranking, while principal component 3 displayed an upward trend in terms of ranking. Among them, the fitting coefficient R-value of principal component 2, which was 0.3722, and ranking were the highest. The fitting coefficient R-value of principal component 3 was 0.1713, and the ranking was the lowest.

**Table 2. Comprehensive scores of top ten high-oleic rape germplasms.**

| Germplasm | First principal component score $F_1$ | Second principal component score $F_2$ | Third principal component score $F_3$ | Fourth principal component score $F_4$ | Comprehensive score $F$ | Rank |
|---|---|---|---|---|---|---|
| 14 | 1211.74 | 1156.94 | -241.71 | -33.67 | 33.49 | 1 |
| 3 | 1422.10 | 851.44 | -252.58 | -11.98 | 32.15 | 2 |
| 5 | 1389.88 | 921.92 | -290.37 | -58.46 | 31.41 | 3 |
| 26 | 1291.59 | 1034.10 | -338.43 | -25.45 | 31.39 | 4 |
| 15 | 1311.58 | 955.149 | -261.51 | -44.53 | 31.37 | 5 |
| 1 | 1308.40 | 1010.03 | -302.18 | -56.34 | 31.36 | 6 |
| 6 | 1379.87 | 980.16 | -323.99 | -78.80 | 31.32 | 7 |
| 9 | 1237.47 | 1045.98 | -249.62 | -77.13 | 31.31 | 8 |
| 4 | 1440.75 | 810.20 | -254.13 | -48.04 | 31.18 | 9 |
| 12 | 1214.59 | 880.18 | -144.84 | -6.89 | 31.09 | 10 |

## Cluster analyses of quality, yield, and resistance traits of different high-oleic rape germplasms

Cluster analyses were carried out after the normalization of 12 personality traits. The results (Fig 5) showed that 30 high-oleic rape germplasms were divided into four categories, and one group was divided into two subclasses. The results showed that the first category included 9 high-oleic rape germplasms, accounting for 30% of the total, which included germplasm Nos. 10, 11, 15, 19, 20, 23, 27, 28, and 29, respectively. The second category included 6 high-oleic rape germplasms, accounting for 20% of the total, which included germplasm Nos. 12, 16, 18, 22, 25, and 30, respectively. The third category included 7 high-oleic rape germplasms, accounting for 23% of the total. It was divided into two subclasses: subclass A included

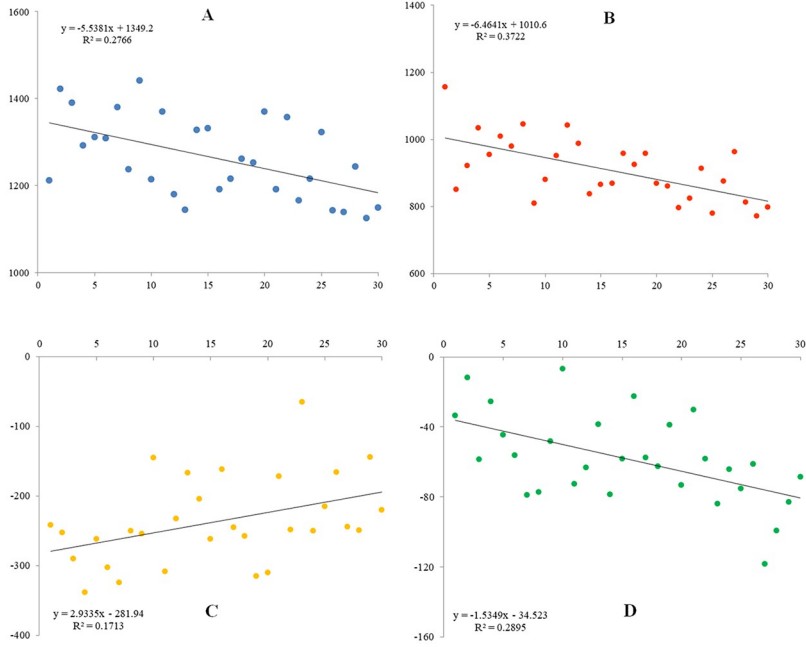

**Fig 4. Trend of comprehensive score.** A: First principal component score; B: Second principal component score; C: Third principal component score; and D: Fourth principal component score. The X-axis is the ranking, and the Y-axis is each principal component.

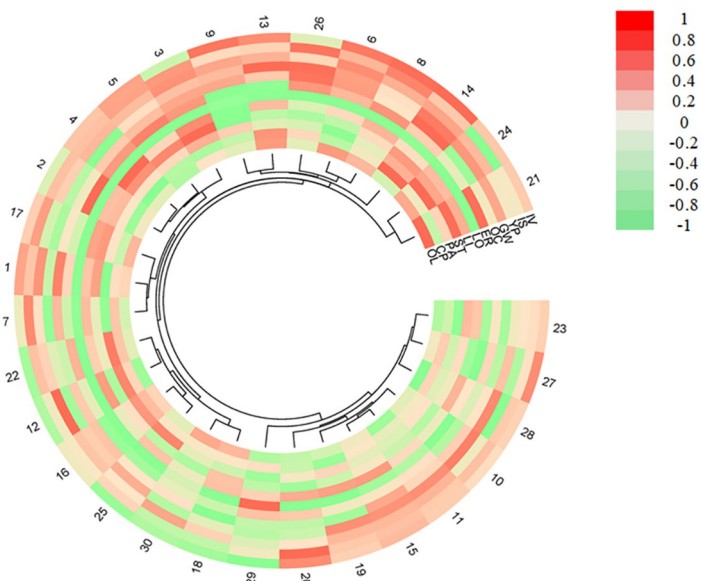

**Fig 5. Hierarchical cluster analyses of traits of different high-oleic rape germplasms.**

germplasm Nos. 1, 7, and 17, while subclass B consisted of germplasm Nos. 2, 3, 4, and 5. The fourth category included 8 high-oleic rape germplasms, which included germplasm Nos. 6, 8, 9, 13, 14, 21, 24, and 26, respectively, accounting for 27% of the total.

The characters of the four groups were calculated, and the results of each character are shown in Table 3. The first group of materials contained erucic acid, but it was still at the "double low" level, and the incidence index was low. The plant yield of the second group was higher than that of other groups, but the incidence rate of *Sclerotium* disease increased. The oil content and crude protein content of this group were high in the third group. The four varieties that have been put into production belonged to this group. Oleic acid content of the fourth group was higher than that of other groups, but the incidence rate of viral disease increased.

## Discussion

### Evaluation and utilization of high-oleic rape resources

*Brassica napus* breeding materials maintain higher genetic diversity due to natural and artificial selection [25]. In recent years, with changes in the orientation of oilseed rape breeding, the

**Table 3. Significance analyses of the differences in the average values of the four groups.**

| Group | OL | CP | PA | ST | LI | LO | ER | OC | GW | YP | IS | IV |
|---|---|---|---|---|---|---|---|---|---|---|---|---|
| I | 80.97 | 21.52 | 2.66 | 1.30 | 4.81 | 2.01 | 0.10 | 42.72 | 3.35 | 21.11 | 15.95 | 1.42 |
| | ab | b | a | a | a | a | a | ab | a | b | bc | a |
| II | 76.83 | 18.35 | 2.94 | 1.10 | 4.93 | 1.88 | 0.02 | 42.56 | 3.35 | 24.64 | 18.95 | 1.76 |
| | c | c | a | a | a | a | a | ab | a | a | a | a |
| III | 80.43 | 23.54 | 3.01 | 0.99 | 6.07 | 2.35 | 0.00 | 44.66 | 3.74 | 21.78 | 13.14 | 1.70 |
| | b | a | a | a | a | a | a | a | a | b | c | a |
| IV | 83.06 | 21.17 | 3.11 | 1.43 | 6.08 | 2.16 | 0.00 | 41.82 | 3.46 | 22.65 | 17.93 | 2.03 |
| | a | b | a | a | a | a | a | b | a | ab | ab | a |

Different lowercase letters in each column indicate significant differences among treatments (p<0.05).

goal has gradually changed to high-quality and multi-purpose varieties. Re-evaluation and utilization of breeding resources is an important way to expand the genetic basis of oilseed rape breeding and make full use of the seed. In addition, the studies on the genetic diversity of quality traits of breeding materials mostly focus on specific types, which is related to the clustering results showing specific groups. The 30 high-oleic rapeseed materials selected in this study showed multiple phenotypes such as "high oil content", "high protein", and "high disease resistance". It was found that the diversity index of other characteristics except erucic acid was less than 1.00. The results showed that the overall changes in the trend of high-oleic rapeseed germplasm resources in the Hunan Province is consistent. However, the standard deviation of some traits such as oil content, protein, and *Sclerotinia* resistance were large, indicating a possibility for further improvement in high-oleic rape breeding.

## Comprehensive evaluation and index screening of phenotypic characters of high-oleic rape germplasm resources

The quality, yield, and resistance traits of rapes are complex quantitative traits affected by their own genetic factors and external environmental conditions [26, 27]. It is difficult to accurately and objectively evaluate the quality of germplasm resources using a single index. In particular, focusing on only the content of oleic acid in high-oleic rapes could result in low yield and serious disease susceptibility. Therefore, this study used multivariate statistical analysis to evaluate high-oleic rape. These methods have been applied in the comprehensive evaluation of rice [28], rape [14], cotton [29], millet [30], and peanut [31]. The evaluation results for this research showed that the approved rape varieties, No.1, 3, and 4 were in the top 10 of the evaluation, indicating that the data statistics were stable and reliable. However, variety No. 2 ranked 22[nd] during our evaluation. Thus, we can speculate that the quality of this variety may have declined slightly after many years of planting because it is a conventional species. Therefore, continuous and uninterrupted breeding of high-oleic rape resource is not only the guarantee of maintaining rape quality but also an assurance of rape yield and resistance.

This study found that the first, second, and fourth principal components showed an upward trend, and the third principal component showed a downward trend in terms of ranking, but the fitting coefficient R-value did not show significance. The reason for this is the limitation of univariate regression simulation. The selection of multivariate or other fitting methods could improve the fitting coefficient. However, even with an increase in the coefficient, the trend cannot change. Therefore, selectively increasing the first, second, and fourth principal components or reducing the traits represented by the third principal component can improve the quality of high-oleic rape varieties. However, it should be noted that there was a saturation range for some traits during selection. For example, the content of erucic acid, the first principal component, was low in this study. An appropriate increase is conducive to the synthesis of fatty acids such as oleic acid, but after reaching a certain degree, this increase may affect the quality. This study can be taken as a breakthrough in the correlation between yield and quality of high-oleic rape, and provide a reference basis for breeding new rape varieties with high yield and improved quality.

This study identified the different types of high oleic acid rape germplasm resources through cluster analysis. The efficiency of breeding can be greatly improved by dividing according to the performance of different traits. Among the four groups, group one can be used to select high oleic acid rape germplasm resources to resist viral diseases. Group two can be used for breeding high oleic acid rape germplasm resources with high yield. Group three can be used as a germplasm resource to breeding high oil content and to resisting *Sclerotinia sclerotiorum*. The content of oleic acid in group four is high, which can be used as a parent for

further cultivation of high oleic acid rape materials. Different group materials can be selected for breeding and improvement according to the actual needs in the future breeding process of high oleic acid rape.

## Correlation and application among quality, yield, and resistance traits

There are correlations among many traits of rape, which can be used as the basis for screening specific germplasm resources. In this study, the correlation analysis of multiple trait indices of high-oleic rape showed that there was a significantly negative correlation between oil and protein content. This was consistent with the results of oil and protein concentration in rape by Delourme [32], and also in agreement with the "substrate competition" hypothesis of Chen Jinqing [33]. At the same time, this study also found a significant positive correlation between oil content and yield per plant. The reason may be that these two traits can control yield and affect each other. Combined with the above views, a rapid method for early and rapid identification of high-oleic rape can be constructed to promote the process of breeding and improvement of high-oleic rape worldwide.

## Conclusions

In summary, the germplasm resources of high-oleic rape in the Hunan Province were relatively stable. In this study, the comprehensive evaluation score of material No.14 is the highest with 33.49, which can be used to declare new varieties and put into actual production in the Hunan Province. However, the properties of other materials need to be further improved. Cluster analysis showed that the germplasm resources could be divided into four categories, with the characteristics of "high disease resistance", "high yield", "high protein", and "more stable" in high-oleic rape. Correlation analysis showed that oil content and yield per plant could be used as indices for early screening of high-oleic rape.

## Supporting information

**S1 Data. The minimal data named "minimal data of 30 high oleic acid rape" has been uploaded in supporting information file.**
(XLSX)

## Author Contributions

**Data curation:** Junjie Wu.

**Formal analysis:** Xuepeng Wu.

**Funding acquisition:** Mei Guan.

**Methodology:** Dongfang Zhao.

**Software:** Mingyao Yao.

**Writing – review & editing:** Tao Chang, Chunyun Guan.

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
