## [Decision Letter · Decision Letter 0]

30 May 2022

PONE-D-22-10737Comprehensive Evaluation of of High-Oleic Rapeseed (Brassica napus) based on Quality, Resistance, and Yield Traits: A new method for rapid identification of high-oleic acid rapeseed germplasmPLOS ONE

Dear Dr. Guan,

Thank you for submitting your manuscript to PLOS ONE. After careful consideration, we feel that it has merit but does not fully meet PLOS ONE’s publication criteria as it currently stands. Therefore, we invite you to submit a revised version of the manuscript that addresses the points raised during the review process.

Both reviewers recommend minor revisions to this study. The topic area is of interest to PLOS One and Brassica community readers. If the revisions are made appropriately, then resubmission to Plos One would be appropriate. I encourage authors to avoid repetitions. Detailed comments are attached. High chlorophyll and erucic acid in rapeseed oil are not desirable traits.

We look forward to receiving your revised manuscript.

Kind regards,

Harsh Raman, Ph.D

Academic Editor

PLOS ONE

Journal Requirements:

"This work was supported by “China Agriculture Research System of MOF and MARA” and “Hunan Agriculture Research System of DARA”.

We note that you have provided funding information. However, funding information should not appear in the Funding section or other areas of your manuscript. We will only publish funding information present in the Funding Statement section of the online submission form. 

"YES - Specify the role(s) played. This work was supported by “China Agriculture Research System of MOF and MARA” and “Hunan Agriculture Research System of DARA”."

7. Please include a copy of Table 4 which you refer to in your text on page 8.

Additional Editor Comments:

Both reviewers recommend minor revisions to this study. The topic area is of interest to PLOS One and Brassica community readers. If the revisions are made appropriately, then resubmission to Plos One would be appropriate. I encourage authors to avoid repetitions. Detailed comments are attached.

Reviewers' comments:

Reviewer's Responses to Questions

**Comments to the Author**

1. Is the manuscript technically sound, and do the data support the conclusions?

Reviewer #1: Yes

Reviewer #2: Partly

2. Has the statistical analysis been performed appropriately and rigorously? 

Reviewer #1: Yes

Reviewer #2: No

3. Have the authors made all data underlying the findings in their manuscript fully available?

Reviewer #1: Yes

Reviewer #2: Yes

4. Is the manuscript presented in an intelligible fashion and written in standard English?

Reviewer #1: Yes

Reviewer #2: Yes

5. Review Comments to the Author

Reviewer #1: The paper is technically sound. There are some minor grammatical errors and some questions regarding results and conclusion. Some of the statements make assumptions that are not based on the data reported. These issues should be addressed prior to publication.

Reviewer #2: This manuscript aims to show a method of statistical evaluation for selection of rapeseed germplasm, although the title doesn't say that explicitly.

The title should be more descriptive of the methodology. The existing title also has grammatical errors.

I have several concerns about the conclusions drawn from the results which I will outlay. But firstly, I am concerned that several references, of which there are an excessive amount, often don't appear to be relevant to the particular discussion. For example, ref. 2-3 is in relation to essential fatty acids but it is in fact about fungicide application. Ref. 5-6 is about cardiovascular benefits but refers to biodiesel and batch reactors? References from 1-31 and others have dubious connections to the related text. The overall number of references can be reduced.

The paper uses 12 indices for the analysis. It is questionable as to the importance of these 12 parameters. Although oleic acid, yield and disease resistance are imported, I am not sure of the need to select for some of the other minor fatty acids. A reduction in traits will significantly improve the clarity of the correlations.

The pages are not numbered so I have presumed the numbering is as they appear on my computer.

Pg. 2, 2nd line, "hit in the throat" needs to be replaced with more acceptable grammar.

Pg. 2, 1st line, 3rd para: should say 'carried out "extensive" work'

Pg 2, 5th para: "During the "decomposition" period" I don't know what the decomposition period is?

Test Methods: It appears 10 plants were selected for analysis but test material consisted of 30 plants. Please explain this better.

Pg 2, last line: rapeseed plants are not "rapes"

Fig. 1: This is hard to read and poorly organised. Improve the labelling and sort the bars so that fatty acids are together, separate from crude protein, etc.

Pg. 5: Fatty acids are percentages of 6 FAs but there are several more. Why are only 6 used to calculate percentage?

Pg. 6, 2nd para: The text is repetitive. You do not need to repeat "This composition mainly reflected ...etc."

Para 3.5: I am not sure of the purpose of this data. The formulas don't seem to help me select new germplasm.

Pages 4-10 show multiple statistical analysis of the data and I don't think it is justified. In fact, Figure 5, hierarchical cluster analysis, is not even discussed within the text. If it is not discussed, or is not necessary, it should be removed. Perhaps the relationships could have been described with less statistical methodology.

The Discussion, page 10-12 shows some limitation to the authors knowledge of rapeseed. Pg 10, 2nd para suggests "previous studies have focussed on yield and not quality" which is totally wrong. The earliest research on Brassica napus was to reduce erucic acid and glucosinolates. From there, oil content was a primary focus. Quality was always a main aim.

It is stated that "for the first time it was shown a negative correlation between oil and protein. This is wrong as correlation between oil and protein has been the subject of multiple publications and scientific presentations for many years. Furthermore, the correlation between chlorophyll and yield is questionable. High concentrations of chlorophyll indicate immature seeds and if allowed to mature, the chlorophyll will disappear. The reasons for green oil are that crops don't reach maturity before they are harvested.

Pg 11: Conclusion. Last line: I don't imagine that "increasing erucic acid to improve yield" would be an acceptable breeding strategy, considering erucic acid is one of the quality limitations of rapeseed.

There are a few grammatical errors:

Pg. 2, para 4: "well-suited to"...;

Pg 2, para 4: "rapeseed oil into high-oleic oil".

There may be others

Overall, I think the manuscript is reasonably well written and with some revision it is publishable. My recommendations are to consider if the wide range of statistical analyses is justified or could the same picture be drawn from a reduced set of select methods? I don't believe the number of references are justified and, where they are used, they must be clearly related to the discussion to which they are referenced. All of the references listed need to be referred to in the text.

I would recommend publication after these factors are dealt with. ality limitations of rapeseed.

6. PLOS authors have the option to publish the peer review history of their article (what does this mean?). If published, this will include your full peer review and any attached files.

Reviewer #1: No

Reviewer #2: **Yes: **Dr R.J. Mailer

---

## [Author Response · Author response to Decision Letter 0]

28 Jun 2022

Dear Editors and Reviewers:

Thank you for your letter and for the reviewer’s comments concering our manuscricpt entitled “Comprehensive Evaluation of High-Oleic Rapeseed (Brassica napus) based on Quality, Resistance, and Yield Traits: A new method for rapid identification of high-oleic acid rapeseed germplasm”. Those comments are all valuable and very helpful for revising and improving our paper, as well as the important guiding ance to our researches. We have studied comments carefully and have made correction which we hope meet with approval. 

The main corrections in the paper and the resp onds to the reviewer’s comments are following:

To Harsh Raman, Ph.D

1. Please ensure that your manuscript meets PLOS ONE's style requirements, including those for file naming. The PLOS ONE style templates can be found at https://journals.plos.org/plosone/s/fil

e?id=wjVg/PLOSOne_formatting_sample_main_body.pdf and https://journals.plos.org/plosone/s/f

ile?id=ba62/PLOSOne_formatting_sample_title_authors_affiliations.pdf.

Reply: This manuscript has been modified according to the style requirements of PLOS ONE.

Reply: ‘Funding Information’ and ‘Financial Disclosure’ have changed to “China Agriculture Research System of MOF and MARA(CARS-13)” and “Hunan Agriculture Research System of DARA(Xiangnongfa[2019]No.105)”.

3. Thank you for stating the following in the Funding Section of your manuscript: “This work was supported by “China Agriculture Research System of MOF and MARA” and “Hunan Agriculture Research System of DARA”. We note that you have provided funding information. However, funding information should not appear in the Funding section or other areas of your manuscript. We will only publish funding information present in the Funding Statement section of the online submission form. Please remove any funding-related text from the manuscript and let us know how you would like to update your Funding Statement. Currently, your Funding Statement reads as follows:“YES - Specify the role(s) played. This work was supported by “China Agriculture Research System of MOF and MARA” and “Hunan Agriculture Research System of DARA”.” Please include your amended statements within your cover letter; we will change the online submission form on your behalf.

Reply: Funding section of manuscript have been deleted. And funding statement don’t need to update.

4. In your Data Availability statement, you have not specified where the minimal data set underlying the results described in your manuscript can be found. PLOS defines a study's minimal data set as the underlying data used to reach the conclusions drawn in the manuscript and any additional data required to replicate the reported study findings in their entirety. All PLOS journals require that the minimal data set be made fully available. For more information about our data policy, please see http://journals.plos.org/plosone/s/data-availability. Upon re-submitting your revised manuscript, please upload your study’s minimal underlying data set as either Supporting Information files or to a stable, public repository and include the relevant URLs, DOIs, or accession numbers within your revised cover letter. For a list of acceptable repositories, please see http://journals.plos.org/plosone/s/data-availability#loc-recommended-repositories. Any potentially identifying patient information must be fully anonymized. Important: If there are ethical or legal restrictions to sharing your data publicly, please explain these restrictions in detail. Please see our guidelines for more information on what we consider unacceptable restrictions to publicly sharing data: http://journals.plos.org/plosone/s/data-availability#loc-unacceptable-data-access-restrictions. Note that it is not acceptable for the authors to be the sole named individuals responsible for ensuring data access. We will update your Data Availability statement to reflect the information you provide in your cover letter.

Reply: Data has been uploaded with the revised version.

Reply: We have no requirement to change.

Reply: Our corresponding author’s ORCID iD is 0000-0003-0175-1423.

7. Please include a copy of Table 4 which you refer to in your text on page 8.

Reply: This is an error. It should be Table 2

Reply: The references have been examined. No withdrawn reference.

Special thanks to you for your good comments.

To Reviewer #1

1.The paper is technically sound. There are some minor grammatical errors and some questions regarding results and conclusion. Some of the statements make assumptions that are not based on the data reported. These issues should be addressed prior to publication.

Reply: Thank you for your comments. The grammatical errors in this paper have been corrected. The questions regarding results and conclusion have been modified according to the reviewers' opinions. Special thanks to you for your good comments.

To Reviewer #2 Dr R.J. Mailer

1. But firstly, I am concerned that several references, of which there are an excessive amount, often don't appear to be relevant to the particular discussion. For example, ref. 2-3 is in relation to essential fatty acids but it is in fact about fungicide application. Ref. 5-6 is about cardiovascular benefits but refers to biodiesel and batch reactors? References from 1-31 and others have dubious connections to the related text. The overall number of references can be reduced.

Reply: References have been deleted and modified.

2. The paper uses 12 indices for the analysis. It is questionable as to the importance of these 12 parameters. Although oleic acid, yield and disease resistance are imported, I am not sure of the need to select for some of the other minor fatty acids. A reduction in traits will significantly improve the clarity of the correlations.

Reply: We think other fatty acids are also necessary. On the one hand, these fatty acids will affect oleic acid metabolism or other traits in the metabolic process. Their analysis will help to speed up the breeding process of high oleic acid rape. On the other hand, a systematic analysis of these fatty acids can provide a reference for the breeding of some specific materials, such as edible high linolenic acid rapeseed oil. Most fatty acid metabolism are interrelated. So we still suggest to keep the original 12 traits.

3. Pg.2, 2nd line, "hit in the throat" needs to be replaced with more acceptable grammar.

Reply: This words has been changed to “affected by”.

4. Pg.2, 1st line, 3rd para: should say 'carried out "extensive" work'

Reply: This word has been changed to “extensive”.

5. Pg 2, 5th para: "During the "decomposition" period" I don't know what the decomposition period is?

Reply: This words has been changed to “experiment”.

6. Test Methods: It appears 10 plants were selected for analysis but test material consisted of 30 plants. Please explain this better.

Reply: The words has been changed to “variety” for better expression.

7. Pg 2, last line: rapeseed plants are not "rapes"

Reply: This word has been changed to rapeseed plants.

8. Fig.1: This is hard to read and poorly organised. Improve the labelling and sort the bars so that fatty acids are together, separate from crude protein, etc.

Reply:This fig has been modified according to the comments.

9. Pg.5: Fatty acids are percentages of 6 FAs but there are several more. Why are only 6 used to calculate percentage?

Reply: The percentage here refers to the proportion of total fatty acids. The six fatty acids here are common traits for screening quality.

10. Pg.6, 2nd para: The text is repetitive. You do not need to repeat "This composition mainly reflected ...etc."

Reply: The text have been deleted.

11. Para 3.5: I am not sure of the purpose of this data. The formulas don't seem to help me select new germplasm.

Reply: The content of this part is that the F value of the comprehensive score of phenotypic traits is used to evaluate the germplasm resources of high oleic acid rape on the basis of principal component analysis. The germplasm resources with higher comprehensive score F value are more suitable for new variety breeding. At present, this method has been applied to many other crops. However, there is no relevant report on the application of high oleic acid rapeseed.

12. Pages 4-10 show multiple statistical analysis of the data and I don't think it is justified. In fact, Figure 5, hierarchical cluster analysis, is not even discussed within the text. If it is not discussed, or is not necessary, it should be removed. Perhaps the relationships could have been described with less statistical methodology.

Reply: We checked these data and corrected the errors. At the same time, the discussion related to cluster analysis was added to make the writing more fluent.

13. The Discussion, page 10-12 shows some limitation to the authors knowledge of rapeseed. Pg 10, 2nd para suggests "previous studies have focussed on yield and not quality" which is totally wrong. The earliest research on Brassica napus was to reduce erucic acid and glucosinolates. From there, oil content was a primary focus. Quality was always a main aim.

Reply: The text have been deleted.

14. It is stated that "for the first time it was shown a negative correlation between oil and protein. This is wrong as correlation between oil and protein has been the subject of multiple publications and scientific presentations for many years. Furthermore, the correlation between chlorophyll and yield is questionable. High concentrations of chlorophyll indicate immature seeds and if allowed to mature, the chlorophyll will disappear. The reasons for green oil are that crops don't reach maturity before they are harvested.

Reply: The words “for the first time” have been deleted. And chlorophyll related parts have been deleted.

15. Pg 11: Conclusion. Last line: I don't imagine that "increasing erucic acid to improve yield" would be an acceptable breeding strategy, considering erucic acid is one of the quality limitations of rapeseed.

Reply: This conclusion has been deleted.

16. There are a few grammatical errors:

Pg. 2, para 4: "well-suited to"...;

Pg 2, para 4: "rapeseed oil into high-oleic oil".

There may be others

Reply: These errors have been corrected.

Special thanks to you for your good comments.

To Comments in the text:

A1. “High-oleic rapeseed oil not only does not smoke when heated to a higher temperature during cooking”-- Change wording to clarify meaning. 

Reply: This sentence has been changed to “High oleic acid rape not only has high smoke point”.

A2. “Since 2020, Guan Chunyun, an academician in”-- Is this necessary?

Reply: This sentence has been deleted.

A3. “rape”-- oilseed rape

Reply: This word has been changed to oilseed rape.

A4. “Previous relevant studies were limited to unilateral character improvement”-- Such as?

Reply: Citation has been added.

A5. “At present, there are no reports on some characteristics of high-oleic rape or their evaluation globally.”-- Repeats the same information as the previous sentence. Consider deleting.

Reply: This sentence has been deleted.

A6. “The variation, correlation, principal component analysis, and cluster analysis of the quality, yield, and resistance traits of high-oleic rape can provide a scientific and effective theoretical basis for breeding high-oleic rape.”-- Repeats the same information as the previous paragraph. Consider deleting.

Reply: This sentence has been deleted.

A7. “excavation”-- Should this be investigation or evaluation?

Reply: This word has been changed to evaluation.

A8. “material”-- Variety?

Reply: This word has been changed to variety.

A9. “and the oil content was measured using the soxhlet method”

Reply:“Soxhlet method” have been changed to “residue weight method”.

A10. “Should consider rewording this paragraph. Only fatty acid composition and protein mentioned to begin with, then oil content, 100-grain weight and yield per plant introduced later. 

Reply: This paragraph has been rewording.

A11. Include units. For example, is protein % of whole seed? Is it as received or at a standardized moisture content? Are fatty acid a % of total fatty acids or just those reported?

Reply: These % units have been explained in this paragraph.

A12. Should the labels in green in the graph show > rather than the < currently shown? For example OL in blue shows 81-84%. The next label should be >84%. Applies to all traits shown on graph. Also, is there a limit of quantification (LOQ) for the fatty acid composition? Rather than reporting ER as 0, should it be reported as < the LOQ for that the laboratory for that analysis?

Reply: The errors in the figure has been modified. LOQ with “double low” rapeseed oil has been added

A13. “Under the influence of double low rapeseed breeding, the coefficient of variation of erucic acid was greater than 1, but the standard deviation was small.”-- Explain what this indicates.

Reply: This sentence has been deleted.

A14. “The results showed that the maximum Sclerotinia sclerotiorum concentration in high-oleic rape was more than three times the minimum value, and the average was 16.42, with a range of 8.3-29.1.”-- Include units

Reply: Units has been added.

A15. “The percentages of oleic, palmitic, stearic, linoleic, linolenic and erucic acid were obtained as percentages of these components in fatty acid. ”-- Explain more clearly. Are these percentages of totaol fatty acids, or just those fatty acids which have been reported?

Reply: This refers to total fatty acids

A16. “‘maintenance breeding’theory”-- Does this need to be explained ?

Reply: This sentence has been deleted.

A17. “In this study, the correlation analysis of multiple trait indices of high-oleic rape showed for the first time that there was a significantly negative correlation between oil and protein content”-- This is a well established correlation in oilseed rape/canola. It is not specific to high oleic rape.

Reply: “For the first time” has been deleted.

A18. “Previous studies have shown a significant correlation between chlorophyll content and yield in the early growth stage of rape, and the chlorophyll content can be efficiently measured by a spectrometer”-- How is this relevant to the current study?

Reply: This sentence has been deleted.

A19. “The highest comprehensive evaluation score of materials was 14 in this study,”-- Clarify this statement

Reply: This sentence has been corrected to "In this study, the comprehensive evaluation score of material No.14 is the highest with 33.49".

Special thanks to you for your good comments.

We tried our best to improve the manuscript and made some changes in the manuscript. We appreciate for Editors/Reviewers’ warm work earnestly, and hope that the correction will meet with approval. Once again, thank you very much for your comments and suggestion.

Best regards

---

## [Decision Letter · Decision Letter 1]

27 Jul 2022

Comprehensive Evaluation of of High-Oleic Rapeseed (Brassica napus) based on Quality, Resistance, and Yield Traits: A new method for rapid identification of high-oleic acid rapeseed germplasm

PONE-D-22-10737R1

Dear Dr. Guan,

We’re pleased to inform you that your manuscript has been judged scientifically suitable for publication and will be formally accepted for publication once it meets all outstanding technical requirements.

Kind regards,

Harsh Raman, Ph.D

Academic Editor

PLOS ONE

Additional Editor Comments (optional):

Thanks for the revisions made

Reviewers' comments:

Reviewer's Responses to Questions

**Comments to the Author**

1. If the authors have adequately addressed your comments raised in a previous round of review and you feel that this manuscript is now acceptable for publication, you may indicate that here to bypass the “Comments to the Author” section, enter your conflict of interest statement in the “Confidential to Editor” section, and submit your "Accept" recommendation.

Reviewer #1: All comments have been addressed

Reviewer #2: All comments have been addressed

2. Is the manuscript technically sound, and do the data support the conclusions?

Reviewer #1: (No Response)

Reviewer #2: Yes

3. Has the statistical analysis been performed appropriately and rigorously? 

Reviewer #1: (No Response)

Reviewer #2: Yes

4. Have the authors made all data underlying the findings in their manuscript fully available?

Reviewer #1: (No Response)

Reviewer #2: Yes

5. Is the manuscript presented in an intelligible fashion and written in standard English?

Reviewer #1: (No Response)

Reviewer #2: Yes

6. Review Comments to the Author

Reviewer #1: (No Response)

Reviewer #2: The authors appear to have complied with reviewer suggestions. I have some minor suggestions for improvement to grammar and presentation:

Line 67-70: I suggest - "This research was conducted in the large rapeseed producing area of Hunan Province. Transforming traditional rapeseed oil into high oleic oil could provide substantial economic and nutritional benefits to 61 million people in the province".

Line 115: replace rapes with rape, rapeseed or rapeseed cultivars.

Line 126: Figure 1 is still not clear on my copy and the numbers on the graph need to be made clearer.

Line 398: Reference 21 needs to be formatted

My suggestions for future research would be to conduct similar studies in variable environments and over successive years of variable climate to see how the relationships hold under greater quality, disease and yield fluctuations.

Additional Comments: The limitations to this work are that the study has been carried out at one site in one year. To look at the characteristics in more detail, and the relationship of one to another (i.e. protein and oil) I would think this needs to be expanded to include multiple years and sites in future studies.

7. PLOS authors have the option to publish the peer review history of their article (what does this mean?). If published, this will include your full peer review and any attached files.

Reviewer #1: No

Reviewer #2: **Yes: **Dr. RJ Mailer

---

## [Editor Report · Acceptance letter]

1 Aug 2022

PONE-D-22-10737R1 

Comprehensive Evaluation of High-Oleic Rapeseed (*Brassica napus*) based on Quality, Resistance, and Yield Traits: A new method for rapid identification of high-oleic acid rapeseed germplasm 

Dear Dr. Guan:

I'm pleased to inform you that your manuscript has been deemed suitable for publication in PLOS ONE. Congratulations! Your manuscript is now with our production department. 

Kind regards, 

on behalf of

Dr. Harsh Raman 

Academic Editor

PLOS ONE